# The Patient-Derived Cancer Organoids: Promises and Challenges as Platforms for Cancer Discovery

**DOI:** 10.3390/cancers14092144

**Published:** 2022-04-25

**Authors:** JuneSung Bae, Yun Sik Choi, Gunsik Cho, Se Jin Jang

**Affiliations:** 1Department of Research and Development, OncoClew Co., Ltd., Seoul 04778, Korea; junesung@oncoclew.com (J.B.); yschoi@oncoclew.com (Y.S.C.); gscho1@oncoclew.com (G.C.); 2Department of Pathology, Asan Medical Center, University of Ulsan College of Medicine, Seoul 05505, Korea; 3Asan Center for Cancer Genome Discovery, Asan Institute for Life Sciences, Seoul 05505, Korea

**Keywords:** patient-derived cancer organoid, drug efficacy test, precision medicine, recapitulation of human cancer biology

## Abstract

**Simple Summary:**

The biopharmaceutical industry increasingly focuses on the development of new anticancer drugs for effective cancer therapy. Despite these efforts, the success rate is low, and thereby, greater efficacy and better safety properties are required. The high failure rate in drug discovery may be due to limitations of preclinical cancer models that inappropriately recapitulate human cancer biology. To date, patient-derived cancer organoids (PDCOs) have emerged as a model system to recapitulate human cancer biology. In this review, we discuss the advantages and applications of PDCO as a model to investigate anticancer drug efficacy and precision medicine. We also describe the challenges that must be overcome so that PDCOs can substantially represent human cancer biology.

**Abstract:**

The cancer burden is rapidly increasing in most countries, and thus, new anticancer drugs for effective cancer therapy must be developed. Cancer model systems that recapitulate the biological processes of human cancers are one of the cores of the drug development process. PDCO has emerged as a unique model that preserves the genetic, physiological, and histologic characteristics of original cancer, including inter- and intratumoral heterogeneities. Due to these advantages, the PCDO model is increasingly investigated for anticancer drug screening and efficacy testing, preclinical patient stratification, and precision medicine for selecting the most effective anticancer therapy for patients. Here, we review the prospects and limitations of PDCO compared to the conventional cancer models. With advances in culture success rates, co-culture systems with the tumor microenvironment, organoid-on-a-chip technology, and automation technology, PDCO will become the most promising model to develop anticancer drugs and precision medicine.

## 1. Introduction

Cancer is the leading cause of death among noninfectious diseases in most countries, and its burden is growing rapidly [1] because of population growth and aging, living environmental pollution, and lifestyle changes [2,3,4]. Despite the advances in cancer prevention and treatment modalities including surgery, radiation therapy, chemotherapy, targeted therapy, and immune therapy, cancer incidence and mortality are still increasing; thus, accelerated programs to develop more effective anticancer therapeutics are required [5,6,7].

In anticancer therapeutic development, well-characterized model systems representing human cancers have been widely used as an essential element. However, the success rate of anticancer drugs was only 3.4% from 2000 to 2015 [8,9,10]. One of the major causes of drug discovery failures is the incompleteness of cancer model systems to recapitulate the biological processes of cancers in the human body. In other words, cancer drug candidates that may be highly developed in the existing in vitro human or in vivo non-human model systems will not be applied to the human body [11,12]. Accordingly, a humanized model system that can provide human physiological characteristics from the development stage of cancer treatment has emerged. Furthermore, drug responses to human cancers highly vary based on an individual patient due to intertumoral heterogeneity. Therefore, newly presented models derived from patients with cancer, including patient-derived cancer cell lines (PDCs), patient-derived xenografts (PDXs), and patient-derived cancer organoids (PDCOs), have been used to discover anticancer drugs. Those patient-derived cancer models have been reported to partly recapitulate human cancers based on cellular heterogeneity (PDC, PDX, and PDCO), drug response (PDC, PDX, and PDCO), and tissue structure (PDX and PDCO) of human cancer [13,14,15].

In this review, we will explore how PDCOs can be used to understand human cancer biology and anticancer therapy, and their advantages over conventional models, such as monolayer culture cells, syngeneic mouse models, and PDXs. Finally, we discuss the challenges faced by the PDCO model and whether it can simulate the physiological and biological properties of human cancer.

## 2. Unmet Needs for the Model System Recapitulating Human Cancer Biology

### 2.1. Non-Human Model System

Model systems for cancer therapeutic development in biomedical sciences range from the monolayer cancer cell culture to animal model systems using *C. elegans*, *D. melanogaster*, and *M. musculus*. The model systems of cancer therapeutics used until the early 2000s primarily aimed to be easily cultured or replicated in the laboratory and had fast-growing properties, allowing them to be built at a relatively low cost. Using these model systems, the classic cancer driver genes, cancer-associated signaling pathways, and therapeutic targets have been identified [16,17]. Moreover, functional properties of certain genes in several species have been reported to be evolutionarily conserved by similar genes in other species [18,19], and non-human origin model systems have been widely used to discover the biological process of human cancer. However, non-human model systems had an essential limitation, mimicking the human physiology in association with the lifespan, metabolism, and immune systems, which were not suitable as a model for developing anticancer drugs.

### 2.2. Patient-Derived Cancer Cell Lines

PDCs can be relatively easily established without advanced techniques, with a short establishment time and the ability to maintain the genetic variation of primary tumors, and the use of large numbers of PDCs can represent intertumoral heterogeneity of cancer patient populations [13]. However, cancer cells cultured using a two-dimensional (2D) monolayer culture method have a major disadvantage because their characteristics may change depending on the culture environment and period [20,21]. PDCs hardly represent features of the original human tumor, such as stromal, immune, and inflammatory cells interacting with tumor cells [22,23,24]. During the long-term culture of PDCs, it can be difficult to maintain as a biobank, as the heterogeneity within tumor cells may disappear, and conventional cancer cell lines and tumor cell senescence may appear [25,26,27,28].

### 2.3. Syngeneic Mouse Models and PDXs

In vitro methods of monolayer cell cultures to investigate anticancer drug discovery are particularly useful for genetic modification and high-throughput screening assays; however, these methods have some limitations, such as the lack of tumor microenvironment. Syngeneic mouse models are thought to overcome some of these limitations and preserve key characteristics of the host tumor, including histological features, tumor microenvironment, and host immune system. In syngeneic mouse models, immortalized murine tumor cells are transplanted into inbred mice with the same genetic background. Allografted murine tumor cells and immunocompetent host mice are generally non-immunogenic, allowing the evaluation of immunotherapy with high reproducibility [29]. However, using murine tumor cells with a murine immune system is limited because they do not accurately recapitulate human cancer biology.

PDXs drew attention as a technology that supplemented the shortcomings of the aforementioned 2D monolayer culture and syngeneic mouse models. Direct transplantation of surgically obtained primary tumor tissues into an immunodeficient mouse allows tumor cells to interact with the microenvironment and to preserve genomic diversity among patients and intratumoral heterogeneity, creating a situation representative of human cancer biology [14]. Although PDX is an excellent animal model for anticancer drug research, several limitations should still be considered. Compared to in vitro models, PDX requires a high cost, long period, and large space to construct and maintain the model. The genetic composition of PDX tumors differs from the original primary tumor due to mouse-specific selection pressures during the establishment period [30,31,32,33,34]. The lack of functional elements of the immune system in immunodeficient host mice for PDX tumor has also the problem that the general PDX model is an unsuitable immunotherapeutic agent evaluation method [35].

These existing model systems are not perfect preclinical models to develop anticancer drugs because they lack the parts necessary to recapitulate human cancer. Given the high cost and long duration of clinical development of anticancer drugs, new and more effective preclinical platforms should be developed for antitumor compound screening. Recently, epithelial cells derived from adult human organs, such as the colon and lungs, have been cultured in a three-dimensional (3D) condition using extracellular matrix (ECM) components to be organized into specific tissue architecture known as the organoids. The ex vivo culture model was adapted for the 3D culture of cancer tissues of a patient, preserving the tissue architecture and genetic characteristics of original human cancers. These patient-derived tumor organoids (PDCOs) models have been proposed as a new alternative model to mimic human cancer biology. Since each model system has pros and cons, the comparison between PDCO and conventional cancer models is summarized in Figure 1.

## 3. Possible Applications Using PDCOs for Cancer Research

Organoids are generally formed through lineage development and differentiation from stem cells (adult, embryonic, and induced pluripotent stem cells) and have the characteristics of self-renewal and self-organization [36,37,38,39,40,41,42]. They can be formed from biopsies directly isolated from diseased patient tissues, including cancer, and researchers have succeeded in culturing cancer organoids from various cancer types [43,44,45,46,47,48,49,50,51,52,53,54,55]. These PDCOs mimic tissue architecture as well as gene expression profiles and genomic alterations of primary cancers, and these properties remain stable in long-term cultures [43,44,45,46,47,48,49,50,51,52,53,54,55]. PCDO is evaluated as an in vitro human cancer model system that can efficiently conduct studies to evaluate anticancer drug efficacy. Previous co-clinical studies have reported that the drug response of PDCO can accurately reflect the clinical outcome of patients with PDCO-matched cancer [56,57,58]. The PDCO characteristics are illustrated in Figure 2.

### 3.1. PDCO Biobank

Compared to PDX, PDCO can efficiently propagate tumor cells ex vivo in a short time with little cost and effort. Additionally, a PDCO biobank can be established from individual PDCOs classified based on the clinical criteria and/or genetic and mutational profiles. Then, it is possible to validate therapeutic agents in PDCOs with specific genomic characteristics and clinical information. Recently, efforts have been made to build such a living PDCO biobank. Corro et al. [11] and LeSavage et al. [59] reviewed the current status of PDCO biobanks built for different cancer types in each organ. PDCOs stored in the biobank are expected to be very useful not only to discover and evaluate an anticancer drug but also for the developing of precision medicine tailored to the patient (Figure 3). Accumulation of drug sensitivity data in PDCOs with known clinical and genomic information can be used to discover new cancer drugs to meet unmet clinical needs [51], predict patient clinical outcomes, and design drug combination therapeutic strategies [50,58,60].

### 3.2. Drug Efficacy Study and High-Throughput Screening

PDCs and animal models are limited in their use to predict patient drug response because these models cannot fully mimic human cancer biology, are costly to use, and have animal ethics issues [61,62,63]. Conversely, PDCOs have not only represented the primary tumor but are also relatively simple culture methods and they avoid ethical issues, making them a suitable model for drug screening. Multiple studies have shown that a preclinical model PDCO has the potential to predict patient drug response in high-throughput screening systems [50,51,58,60,64,65]. The PDCO model is further developed to reduce the labor power and time consumption required for high-throughput screening. The ordinary organoid culture, in which dissociated cells and sticky ECM hydrogel are mixed and dispensed and a culture medium is added, is more difficult and time-consuming than the culture of 2D monolayer cells. The liquid handling robotic systems can culture large amounts of PDCO with smaller efforts and can minimize inter- and intra-personal variability, enabling drug screening with high reproducibility [66,67,68]. Furthermore, large-scale drug response data analysis can be automated by measuring image data, such as changes in the organoid size or morphologic characteristics of PDCO after drug treatment [69,70,71]. The ultimate goal in this field is to automate the entire high-throughput screening process, including organoid cultures, seeding of dissociated cells, drug treatment, and drug response analysis. With this physiologically relevant cancer model, automated culture and large-scale data analysis on the high-throughput platforms will be effectively used to identify novel drugs for cancer treatment.

### 3.3. Precision Cancer Medicine

Because a measurable amount of PDCO can be cultured in a short time from a small tumor sample, it is a well-suited model for screening drugs to be administered. In the PDCO model, the patient’s drug response can be predicted within 3 months, depending on the tumor sample types or the number of drugs to be tested [72]. Several studies have shown that drug screening results using PDCO are very similar to patient clinical outcomes [49,50,54,58,65,72,73,74]. The genetic information of PDCOs is also a powerful tool to predict patient clinical outcomes. For example, breast cancer organoids with the BRCA1/2-associated mutational signature were highly responsive to poly(ADP-ribose)polymerase inhibitors (PARPi). Conversely, the immunohistological analysis of protein biomarkers for tumor samples, a basis of the current precision medicine, did not completely match the drug response. Drugs targeting the HER signaling pathway in breast cancer organoids may not exhibit the expected reactivity depending on the immunological staining status of HER [50]. Therefore, the PDCO model, which accurately reproduces the genetic profile and drug responsiveness of a patient’s primary tumor, is expected to be actively used for cancer medicine with the current pathological biomarker analysis of tumors.

## 4. PDCO Challenges as Cancer Biology Models

Some challenges to improving the understanding of human cancer biology and the success rate of drug discovery in PDCO models remain to be elucidated. The success rate of establishing PDCO remains to be increased. The co-culture methods with PDCO and cellular components of the tumor microenvironment to represent a patient’s tumor tissues have not yet been standardized. Currently used supporting systems for PDCO culture, including the mouse sarcoma-derived matrix and artificial hydrogel, differ from the ECM of primary tumors.

### 4.1. Establishment Rate of PDCOs

One of the unresolved problems in cancer studies using cells derived from patient tumors is that not all patient-derived tumor cells are not successfully cultured. This problem is considered a fatal flaw in all PDC, PDX, and PDCO systems. Moreover, Kodack et al. reported that the success rate of 568 PDC lines was approximately 26%, and the success rate varies based on the tissues from the originating tumor. In the case of lung cancer, 109 PDC lines were successful in 373 tumor tissues, and the success rate was 29%. For breast cancer, only 16 PDC lines of 104 tumor tissues were successfully established with a proportion of 15% [75]. The causes of culture failure were lack of cancer cells after tumor tissue dissociation, excessive growth of stromal cells, or non-growing cancer cells. To increase the success rate of PDC line construction, the medium composition was changed, and irradiated fibroblast feeder cells were used; however, no statistically significant increase in the success rate was observed. PDX models also show varying engraftments depending on the types, origins, and characteristics of the primary tumor [76,77,78]. The engraftment rate greatly varies depending on primary tumor characteristics used for xenograft. In a low tumor burden or stage, the success rate of constructing the PDX model is also low [79,80]. Conversely, metastatic cancer shows a very high engraftment rate in the PDX model [81,82,83,84,85,86]. The difficulty of establishing a non-aggressive primary tumor as a PDX model is a factor hindering the clinical application of the model for precision cancer therapy.

Although the culture success rate of PDCO models is higher than that of PDC models, the success rate also differs depending on the tumor type [87]. For example, the success rate of establishing lung PDCO varies among different studies from 41% to 88% [44,45,88,89,90,91,92,93,94,95,96,97]. In our experiences, the success rates depend on quality control for the following culture steps: initial tumor quality check, early organoid formation after seeding, organoid expansion by culture with proper media, control of fibroblast overgrowth, control of overgrowth of normal organoid, success of long-term culture, freeze–thaw cycle, etc. The factors determining the success rate of PDCO culture have not yet been fully elucidated. Tumor cellularity of the resected primary tumor tissue, culture medium composition, normal epithelial cell contamination, and limited tumor microenvironments have been suggested as causes [68,90,98,99].

### 4.2. Requirements to Recapitulate Human Cancer Biology

Tumor progression is frequently accompanied by expanding neoplastic cells, ECM remodeling, and active interactions with stromal/immune cells [100]. Oxygen and nutrients required for tumor growth depend on the blood supply, and angiogenic factors derived from cancer cells induce angiogenesis to promote tumor growth and metastasis [101,102]. To represent human cancer biology, the most current PDCO models have limitations that should be overcome: ECM composition different from in vivo, interaction with vasculature and tumor microenvironmental cells, and biological knowledge of the different media compositions required for various PDCOs with diverse genetic profiles.

#### 4.2.1. Matrix

The ECM has been found to be a necessary element for reproducing the properties of living tissues in vitro. An ECM matrix derived from Engelbreth–Holm–Swarm (EHS) mouse sarcoma has been shown to be important for the formation of 3D ducts and lumens in the mammary epithelium and maintaining the differentiation of alveolar type II epithelial cells [103,104]. Tumor cells are affected by biochemical and biophysical signals in the microenvironment matrix [105]. Unlike healthy tissues, ECM surrounding tumors have altered composition, architecture, and biological characteristics, which are known to affect tumor progression, tumor metastasis, and drug response [106,107]. Therefore, mechano-transduction and interaction between tumor cells and ECM are expected to be utilized as new drug targets [108]. However, the matrices used for most organoid cultures, including PDCO, are basement membrane preparations derived from EHS mouse sarcoma and do not mimic human tumor and ECM interactions. As the EHS matrix does not have the same biochemical and mechanical properties as the stroma of primary tumors, the mechanism of its effects on PDCO is unknown [109,110]. Furthermore, the EHS matrix has become a major factor unfavorable to PDCO culture due to its high cost.

#### 4.2.2. Vasculature

Tumor blood vessels are known not only to supply nutrients to tumor cells but also to interfere with T-cell-mediated immune surveillance and induce immune evasion of cancer cells [102]. Therefore, antiangiogenic therapies targeting VEGF or EGF have been attempted for cancer treatments [111,112]. Although these attempts have not yet achieved successful clinical results, antiangiogenic therapies, either alone or combined with immunotherapy, are still considered an attractive way to inhibit tumor growth [113]. The drug distribution, i.e., a factor associated with drug effects on cancer cells in solid tumor tissues, depends on plasma pharmacokinetics, tumor vasculature, and its function [114,115]. Simply treating a drug in the culture medium of PDCO that does not contain blood vessels differs from drug delivery in vivo, where it utilizes the bloodstream as its main transporting route and crosses the vascular endothelium before affecting the cancer cells [116].

#### 4.2.3. Stromal, Immune Cells

The tumor microenvironment consists of several types of nontumor cells, including cancer-associated fibroblasts (CAFs), endothelial cells, pericytes, immune cells (lymphocytes, neutrophils, dendritic cells, and monocytes), myeloid-derived suppressor cells (MDSCs), mesenchymal stromal cells, and platelets [117,118,119]. These stromal cells are known to be implicated in cancer biology by interacting with tumor cells that communicate through chemokine, growth factor, enzyme, and extracellular vesicle expressions [120]. For example, normal fibroblasts are activated by tumor-associated factors, including inflammatory signals, physical changes of ECM, and DNA damage responses. CAF activation is involved in tumor progression and metastasis through matrix remodeling, altering the metabolic status of tumors, and modulating immune cells through soluble factor secretion [121,122]. Therefore, tumor-associated stromal cells constituting the tumor microenvironment to the PDCO model system should be identified to understand cancer biology or discover anticancer drugs.

### 4.3. Advantages of PDCO Models

The ultimate goal of the PDCO model is to create an in vitro system capable of mimicking the in vivo tumor microenvironment and recapitulating the original tumor characteristics, i.e., indicative of human cancer physiology. The genetic profiles and intratumoral heterogeneity of primary tumors are preserved in PDCO. Furthermore, well-established PDCO biobanks can reveal intertumoral heterogeneity of individual cancer patient populations. For the PDCO cultivation, a study of matrices and soluble factors in the media to maintain the original tumor properties and co-culture system of tumor organoids and microenvironments mimicking those of primary tumors will be another opportunity (Figure 4).

#### 4.3.1. Media

The stem cell niche factors supplementing the media are very important to expanding normal organoids [123]. In this section, we will review the medium composition for culturing lung PDCO and explore the elements required for the growth and maintenance of the lung stem cell niche and PDCO.

Potential adult epithelial stem cell niches have been identified in the airway and alveoli of lung [124]. Activation of WNT/β-catenin signaling, inhibition of TGFβ/BMP signaling, and FGFR stimulation are important for maintaining stemness of basal cells and alveolar type Ⅱ cells, which differentiate toward airway epithelial cells (ciliated, secretory, columnar cells) and alveolar type I cells, respectively [125]. Identified niche factors provide a way to establish the lung airway and alveolar organoids (Table 1). The difference in media composition between lung airway and alveolar organoids is the presence of WNT3A, which initiates WNT/β-catenin signaling [45,126,127]. Airway organoids with elevated WNT3A levels do not require the addition of exogenous WNT3a [45]. However, stimulation and amplification of WNT/β-catenin by GSK3 inhibitor or R-spondin 1 is important to long-term expansion of airway organoids as well as alveolar organoids [45,126,127,128,129]. TGFβ and BMP signaling should be inhibited for maintaining stem cells and blocking their differentiation by Noggin and ALK5 inhibitor, respectively [126,128,129,130,131]. FGFR ligand is a crucial factor for organoid survival and differentiation to distal lung lineages [127,128,129]. For successful establishment of lung, organoids need additional supplements such as the EGF ligand, which drives proliferation of organoids, and p38 MAPK inhibitor, which protects cells from environmental-stress-induced apoptosis [45,126]. Unlike normal epithelial organoids, PDCOs may not require the same specific niche factors as normal organoids due to genetic alteration. To prevent contamination of normal epithelial organoids, specific niche factors are commonly excluded, resulting in robust expansion of PDCO alone [132,133]. For example, >90% of colorectal cancers contain mutations in at least one of the proteins constituting the WNT/β-catenin signaling pathway and increasing the transcription of cancer-associated genes [134]. The majority of colon cancer organoids can robustly proliferate even when WNT/β-catenin signaling pathway activators, such as WNT3a or R-spondin1, are removed from the culture medium [135]. For lung cancer organoids, selective propagation of TP53-mutated lung PDCOs could be stimulated by senescence or apoptosis induction of TP53 wild-type normal airway organoids after adding Nutlin-3a to the medium [45]. The composition of the culture medium for lung PDCO is largely divided into a culture medium containing WNT activator, or not (Table 2). Overall, presence and absence of WNT activator in culture medium does not seem to affect the organoid establishment rate [44,45,88,89,90,91,92,93,94,95,96,97]. Similar to a normal airway organoid, which is not dependent on WNT3a ligand for organoid formation [45], the activation of WNT pathway is reported in around 50% of non-small-cell lung cancers (NSCLCs), which is the most common primary lung cancer [136]. Dvl-3, a critical mediator of WNT signaling, is overexpressed in 75% of NSCLCs [137]. Recently, expression of Porcupine, an enzyme required for WNT secretion, has driven the proliferation and progression in lung adenocarcinoma [89,138]. Thus, WNT activation is not essential for lung PDCO culture. For example, Kim et al. established 20 lung PDCOs from 23 epithelial-cell-predominant patient’s samples using only EGF- and FGF2-supplemented basal medium [44]. Similar composition of medium was used to negatively select normal lung organoids and the tumor cell percentages increased by about 1.6-fold [96]. Although lung PDCOs containing intra-tumoral WNT-producing and responding cells may not need the WNT activator for short-term culture, WNT ligand or amplification is required for the long-term expansion of lung PDCOs. Choi et al. reported that the addition of WNT3a or R-spondin1, as well as ALK inhibitor and BMP inhibitor, is a prerequisite for the long-term expansion of small-cell lung cancer (SCLC) tumor organoids [95].

Besides the most important components for the culture of lung PDCOs, such as WNT3a, R-spondin, Noggin, FGF2/7/10, and EGF, other factors are also added to the culture medium for growth or maintenance of lung PDCO (Table 2). Li et al. used Gastrin 1 for the culture of lung PDCOs [88]. Gastrin receptors, CCKAR and CCKBR, are abundantly expressed in lung cancer [139,140], and CCKBR antagonist inhibits SCLC cell proliferation [141]. The activation of CCK receptor causes elevation of cytosolic Ca2+, which leads to PKC-dependent Src phosphorylation in SCLC cells [142]. Activated MMP by phosphorylated Src cleaves proTGFα to active TGFα, which binds to the EGFR [141,143]. Therefore, the addition of gastrin to the lung PDCO culture can help the growth and survival of cancer cells by TGFα-EGFR signaling activation.

Sonic hedgehog (Shh) signaling is predominantly activated in both NSCLC and SCLC tissue sample [144,145,146] and the inhibition of Smoothened (Smo), which is a key transmembrane protein involved in transduction of Hh signal into the cell, inducing inhibition of tumor growth [146,147]. In NSCLC, cancer stem cells (CSCs) produce full-length Shh proteins that promote proliferation of cancer cells in a paracrine manner [148]. Since the Hh signaling may also regulate lung stem cell niche, Smo agonist (SAG) is considered an important factor for lung PDCO culture [93].

Prostaglandin E2 (PGE2) and Forskolin are supplemented in the culture medium for lung PDCOs [88,91,96]. PGE2-bounded PGE2 receptor subtype EP4 stimulates normal bronchial epithelial cell growth through induction of PDK1 [149] and PGE2-EP1 receptor, inducing tumor cell migration and metastasis in NSCLC through induction of β1 integrin expression [150,151]. Forskolin activates the adenylate cyclase activator and increases intracellular levels of cAMP [152]. cAMP-PKA-CREB signaling is significantly upregulated compared with normal tissue in lung cancer, and the inhibition of CREB activity abolishes the development of SCLC [153,154,155]. Interestingly, PGE2 induces the expression of COX2, which leads to an increased abundance of PGE2, via activation of PKA-CREB signaling [156]. The significant abundance of PGE2, CREB, and COX2 in lung cancer and their positive feedback loop may play an important role in modulating cytokine balance and lung carcinogenesis [157,158,159].

Neuregulin 1 (NRG1), also referred to as Heregulin-β1, is a ligand for HER3 and HER4, and is aberrantly overexpressed in NSCLC [160,161]. Generally, NRG1 activates HER3, HER4, and its coupling partner HER2, leading to activation of MAPK/ERK, PI3K/AKT/MTOR, JAK/STAT, and PKC, which promote proliferation and migration fo cancer cells [162]. It has been reported that the NRG1 was used for the culture of lung PDCOs [89,97]. Similarly, insulin-like growth factor 1 (IGF1) initiates multiple signaling pathways including PI3K/AKT, MAPK, JAK/STAT, Src, and FAK, which stimulate cancer cell growth [163]. Activation of c-Met, whose only known ligand is HGF, is also important in downstream effectors through the RAS/MAPK, PI3K/AKT, integrins signaling [164,165]. A high plasma level of IGF1 and an amplification or overexpression of c-Met protein is commonly found in SCLC and NSCLC, respectively [166,167,168]. Therefore, IGF1 and HGF are also considered as supplements for lung PDCO culture [91,97].

Activin A (ActA), known as the TGFβ superfamily, shows increased expression levels and stimulates cell growth and proliferation in lung adenocarcinoma [169,170]. ActA is capable of inducing and maintaining mesenchymal phenotype of cancer-initiating cells and promoting lung metastasis via NF-κB activity [171]. Interestingly, stromal fibroblasts-derived IL-6, ActA, and G-CSF drive dedifferentiation of lung carcinoma cells into CSCs under nutrients and oxygen deprivation [172]. Endo et al. used ActA containing serum-free medium for the culture of cancer tissue-originated spheroids. Their spheroids include some cells with characteristics of CSCs, and the expression of the CSC marker CD133 was similar in a way to those in the original tumors [97]. The culture of PDCOs utilizing the characteristics of CSC showed that PDCOs could be maintained even in the absence of essential niche factors such as WNT activator.

However, intra- and intertumoral heterogeneities of tumors suggest that a standardized medium composition cannot be determined for each cancer type. Therefore, specific culture medium composition should be developed for culturing a specific tumor based on genomic, transcriptomic, or proteomic tumor properties [173,174].

#### 4.3.2. Matrix

Currently, many synthetic materials are being developed to replace animal-cell-derived matrix in induced pluripotent stem cells (iPSCs)-derived normal organoids or cancer cell lines culture [59,175]. To replicate key features of the tumor microenvironment, encapsulation of tumor cells within multiple biomaterials have been extensively studied [175]. Biomaterials for 3D culture of cancer cells are alginate [176,177,178,179,180,181], collagen [182,183], gelatin [184,185,186,187,188], polyethylene glycol (PEG) [189,190,191], hyaluronic acid [192], and chitosan [193]. These studies show that the biomaterials can replace the tumor microenvironment and are suitable for the investigation of cancer biology or to confirm the efficacy of anticancer drugs in 3D culture models. Since these biomaterials are freely tunable factors, it is possible to change the composition, concentration, or mix them with other biomaterials to culture various cancer cells. The 3D culture model of cancer cells using biomaterials makes it possible to recapitulate in vivo tumor stiffness [177,179], co-culture with stromal cells [178,181,184,185,186,187,192], and visualize the in vitro cancer invasion [182,184]. However, the majority of studies have been conducted on the well-established cancer cell line, such as MCF7, MCF10, and MDA-MB-231. Since PDCOs cannot be maintained in a 2D culture, it is difficult for many biomaterials to be used for PDCO culture. Additionally, the gelation of the most synthetic biomaterials is relatively complicated compared to the gelation of EHS matrix and there is no standardized method for the subculture of the generated organoids. In the PDCO model, culturing glioblastoma organoids was successful in a polyethylene glycol-based synthetic matrix and human pancreatic ductal adenocarcinoma organoids in fibrin supplemented with laminin [194,195]. Developing synthetic ECM that fully mimics in vivo tumors remains challenging, and ECM factors involved in tumor progression have not been identified. The current research focused on developing synthetic materials capable of culturing multiple organoids and evaluating factors modulating the organoid physiology [59]. For example, PEG-based synthetic materials have been used for the growth and differentiation of intestinal and endometrial organoids, and recombinant hyaluronic acid contents of the synthetic matrix affect the drug response of PDCOs [195,196]. Culturing organoids in tunable matrices can serve as an ideal model to determine ECM components to construct a complete in vitro matrix for each organ or tumor type. Unlike the PDX model, key elements of PDCO culture, such as the ECM and soluble factors, can be altered. These findings indicate that the PDCO model is more suitable to construct systems that mimic the tumor and tumor TME of each patient and is expected to be further developed with biomaterials in the future.

#### 4.3.3. Vascularization

Recent studies applied the vasculature, which is closely associated with tumor progression and drug response, in a PDCO model. First, using the angiogenesis model for PDCO culture, blood vessel organoids were generated from pluripotent stem cells [197] and formed vessel-like structures in cerebral organoids [198,199]. Second, the vasculature was simulated using artificial structures. The flow direction of the culture medium in this system is determined from the inlet to the outlet, and the human living cells, including PDCOs, are cultured around an artificial vessel-like structure [200,201,202,203]. Third, a complicated structure mimicking living tissues was generated using 3D bioprinting with materials known as “bioink” [204,205,206,207]. The 3D bioprinting has the advantage of the ability to customize the shapes, sizes, and various TMEs of PDCOs because the amount of printed bioink can be controlled and living cells included in the bioink can be selected in various manners. Moreover, breast tumor organoids can be formed using 3D bioprinting technology [208], and the culture medium can be also perfused by creating a simple vascular channel inside the matrix in which organoids are aggregated [209]. The challenge facing this field is how to apply delicate vasculatures to the organoids smaller than actual organs and standardize the formation of structurally and functionally abnormal tumor vasculatures [100,102,210]. If these problems are resolved through further development, the PDCO model will accurately identify the primary tumor and is expected to be widely used in preclinical studies that can test antiangiogenic agents.

#### 4.3.4. Immune and Stromal Cells

In the PDCO model, co-culture with various human living cells, including immune cells, has been attempted. PDCO co-culture with patient-specific tumor-reactive T cells allowed us to evaluate the efficacy of immunotherapy against different tumors [211,212,213]. In a recent study, MDSCs, known to block the immune response of CD8+ T cells, inhibit T-cell cytotoxicity in PDCO co-culture with immune cells [214].

When PDCOs were co-cultured with CAFs and immune cells via an air–liquid interface, the stromal composition and immune repertoire were similar to those of the original tumors. PDCOs have successfully evaluated the immune checkpoint blockade (ICB) and confirmed tumor cytotoxicity [215]. Chalabi et al. suggested that the co-culture method of PDCOs with autologous T cells would help identify the causes of non-responsive neoadjuvant immunotherapy and find a new ICB or targeted therapy method [216]. In other studies, CAFs promoted organoid formation in circulating tumor cells and cancer cell lines [217,218]. Another study reported that liver PDCOs showed resistance to clinically used anticancer drugs when co-cultured with CAF [219]. Blocking IL-6 or ADAM12 secreted by CAF has been reported to inhibit anticancer drug resistance and tumor migration ability in co-culture with esophageal adenocarcinoma-derived PDCOs and CAFs [220].

As the PDCO model provides an ideal environment for co-culture with other nontumor cells, it would be a good model to elucidate the effects of various stromal cells on tumor progression, metastasis, and drug resistance.

#### 4.3.5. Organoids-on-a-Chip

As most in vitro cell culture systems lack vasculature, tissue–tissue interaction, and interaction with stromal/immune cells, these limitations make it impossible to completely recapitulate the human tumor biology. To solve this problem, the “organ-on-a-chip” technology has been developed that can attach and grow various types of human living cells to a microfabricated device [221]. The organ-on-a-chip can be used as a model to mimic human physiological homeostasis and disease progression and thus can be used for preclinical evaluation [222] and precision medicine [223,224]. It is also considered a platform for drug absorption, distribution, metabolism, and excretion analysis [225,226]. Organoids also aim to represent human biology differently; thus, the “organ-on-a-chip” technology that combines characteristics of organ-on-a-chip and organoid is considered a system reflecting human physiology. One of the differences between the general organoid culture method and organ-on-a-chip is the environment for the presence or absence of vascularization. The growth of PDCOs without vasculature relies on limited nutrients in the media inside the dish, whereas in vivo tumors are continuously nourished by perfusable blood. Tumor progression, including cell proliferation, angiogenesis, and cell migration, can be observed, and physiological conditions of the drug delivery through blood vessels can be simulated on an organoids-on-a-chip [227,228,229]. Tumor microenvironment and vasculature are important factors for tumor progression and drug sensitivity [230,231,232]. Recent studies have shown that tumor organoids can be co-cultured with multiple cell types, such as the ECM [233], immune cells [234,235,236], endothelial cells [237], mammary/lung fibroblasts [233,238], and CAFs [239], on a chip. In these studies, the effects of matrix and stromal cell composition on cancer cells [233], immune blockade on tumor treatment efficacy [161,162,163], a relationship between chemokines secreted from stroma and metastasis [237], and effects of the surrounding cells, such as the fibroblasts on drug sensitivity [233,238,239], have been reported. Because the ultimate goal of the organoid-on-a-chip technology is to implement complex human physiology and pathology in vitro, it can be used in drug discovery and precision cancer medicine to guide the treatment of cancer patients and to replace animal tumor models with many limitations associated with the cost, space, and ethics.

## 5. Conclusions and Perspective

All existing cancer model systems have pros and cons when used as platforms for studying cancer biology and anticancer drug discovery. Among them, PDCOs are in a unique position. PDCOs preserve the genetic, physiological, and histologic characteristics of original cancer and accurately predict a patient’s drug response. Individual PDCO models in large cohorts can represent individual cancer patients in a cancer population. With these advantages, the PCDO model is attempted for efficacy testing of new targeted anticancer drugs, phenotype-based drug screening, patient stratification before clinical trials, and anticancer drugs selection for personalized therapy. Although many PDCO models still need to be developed in terms of the success rate, co-culture system with the tumor microenvironment, and organoid-on-a-chip technologies, PDCO will be placed as the most promising model applicable to preclinical and clinical development. Advances in organoid technology over the past decades have made it possible to generate PDCOs from small tumor biopsies. Using microfabricated devices, PDCOs can serve as a platform to predict clinical outcomes before anticancer therapy. Furthermore, a large-scale drug screening using PCDOs with automation technology is expected to dramatically accelerate anticancer drug development. From this point of view, much effort is needed for the development of PDCO technology. To increase the success rate of establishing PDCOs, the medium composition should be optimized for the characteristics of each primary tumor. Standardization in PDCO culture using biochip technology with well-matched media and matrix may facilitate clinical application of the PDCOs. In conclusion, the PDCO model will be a powerful tool to create the most accurate system for precision medicine and to develop innovative therapies for a cancer-free world.

## Figures and Tables

**Figure 1 cancers-14-02144-f001:**
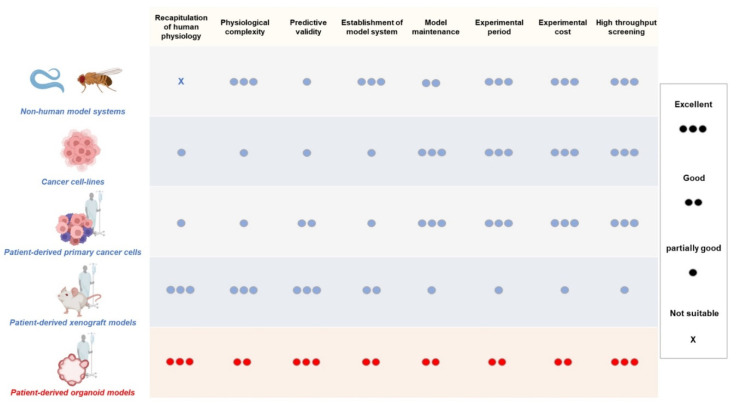
Comparison between PDCOs and existing cancer models. Model systems that are used in developing anticancer drug and drug efficacy study are non-human model systems, cancer cell-lines, and patient-derived xenografts (PDXs), along with patient-derived cancer organoids (PDCOs). Each model system has its own advantages and disadvantages for cancer research. Relative properties are represented as being excellent (three circle), good (two circle), partially good (one circle), and not suitable (cross). Figure was created using BioRender.com (accessed on 31 March 2022).

**Figure 2 cancers-14-02144-f002:**
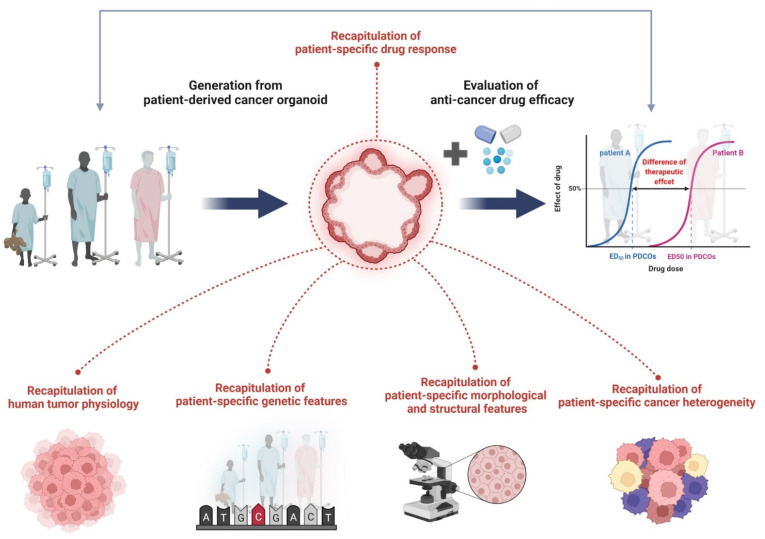
The characteristics of PDCOs in terms of recapitulation of human cancer biology. Organoids derived directly from human tumor tissue have been shown to accurately recapitulate tumor physiology, genetic features, histological properties, and cancer heterogeneity. Specifically, drug responses of PDCOs can exactly reflect clinical outcomes of PDCO-matched cancer patients. As a result, PDCOs are a promising tool to evaluate anticancer drug efficacy studies in clinic. Figure was created using BioRender.com (accessed on 31 March 2022).

**Figure 3 cancers-14-02144-f003:**
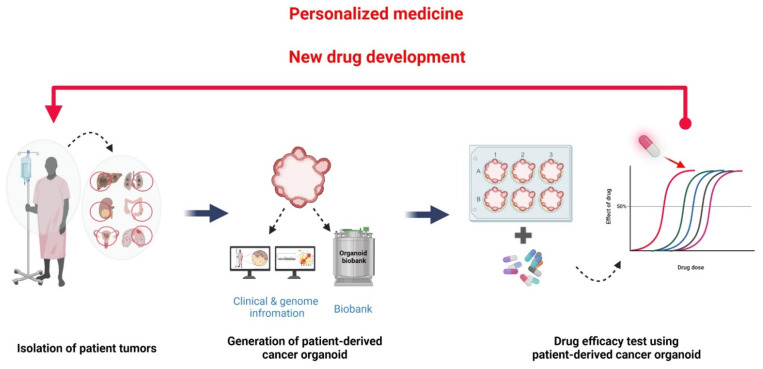
Construction of living PDCO biobank with clinical and genomic information. PDCOs are efficiently propagated from a patient’s tumor biopsy. This property is an advantage for using PDCOs as a living biobank. PDCO biobanks can be classified using a patient’s information, including medical history, cancer type, medication, and surgery. Genomic information, which might inform the utilization of targeted therapies, also can be used as a criterion in PDCOs classification. Discovering the specific drug sensitivities of PDCOs with known clinical and genomic information are essential for the discovery and evaluation of anticancer drugs as well as personalized medicine. Figure was created using BioRender.com (accessed on 31 March 2022).

**Figure 4 cancers-14-02144-f004:**
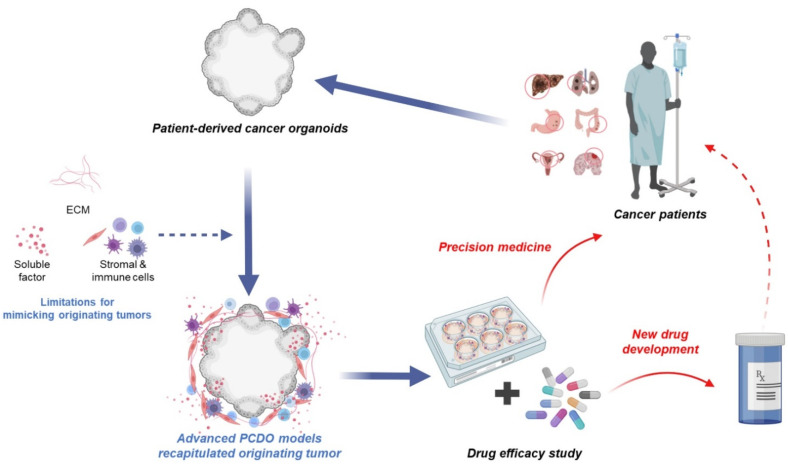
Challenges and opportunities of the PDCOs model. Cancer comprises a collection of abnormally proliferative cells growing within a tumor microenvironment. The dynamic interactions of cancer cells with their microenvironment consisting of stromal cells, ECMs, extracellular ligands, and vasculature is essential to tumor progression and anticancer drug response. Therefore, tumor microenvironments are important considerations for PDCOs culture. Further development of a co-culture system with PDCOs and tumor microenvironment will recapitulate complex human physiology and pathology in vitro. In addition, it would be used for precision medicine and drug discovery to treat patient-specific tumors by replacing existing models. Figure was created using BioRender.com (accessed on 31 March 2022).

**Table 1 cancers-14-02144-t001:** Niche factors and supplement for lung normal organoids.

Niche Factor	Supplement	Role	Functions	Ref.
WNT	WNT3a	WNT ligand	initiates the canonical Wnt/β-catenin pathway. WNT3a is required to the clonal expansion of Alveolar type 2 cells Airway organoid is not required the addition of exogenous WNT3a.	[45,117,118]
	CHIR99021	GSK3 Inhibitor	stimulate the WNT/β-catenin signaling strong wnt signaling impairs the formation of airwary organoid	[119,120]
	R-spondin	agonist	amplifies the WNT/β-catenin signaling. require the expansion and long-term culture of airway organoid.	[45,117,118]
BMP	Noggin	BMP inhibitor	increases the number of stem cells and blocks their differentiation. maintains the stemness of stem cells.	[112,117,119]
TGF-b	SB431542	ALK5 inhibitor	lengthen organoid growth	[119,120]
	A83-01	ALK4/5/7 inhibitor	prevents TGF-β induced growth inhibition.	[121]
FGF	FGF2	FGFR ligand	keeps the survival of organoids. produce large organoids and induce organoid branching	[118]
	FGF7	FGFR ligand	promotes differentiation of lung stem cells toward distal lung lineages. induces organoid branching.	[118,119,120]
	FGF10	FGFR ligand	promotes differentiation of lung stem cells toward distal lung lineages. induces organoid branching.	[118,120]
EGF	EGF	EGFR ligand	drive proliferation of organoids. not essential for organoid formation, but it increases the size of alveolar organoids	[117]
p38 MAPK	SB202190	p38 MAPK inhibitor	to overcome organoid growth arrest protects cells from environmental stress induced apoptosis.	[45]

**Table 2 cancers-14-02144-t002:** Media composition of lung PDCOs.

		Li et al. [127]	S´andor et al. [128]	Dijkstra et al. [90]	Taverna et al. [129]	Li et al. [130]	Shi et al. [131]	Sachs et al. [45]	Kim et al. [132]	Choi et al. [133]	Kim et al. [44]	Hu et al. [134]	Endo et al. [135]
WNT	WNT3A	100 ng/mL	-	-	-	-	-	-	-	0 or 100 ng/mL	-	-	-
	CHIR99021	-	-	-	-	-	250 nM	-	-	-	-	-	-
	R-spondin	250 ng/mL	500 ng/mL	10%	500 ng/mL	500 ng/mL	-	500 ng/mL	20%	0 or 10%	-	-	-
BMP	Noggin	100 ng/mL	100 ng/mL	10%	100 ng/mL	100 ng/mL	100 ng/mL	100 ng/mL	100 ng/mL	0 or 100 ng/mL	-	-	-
TGF-b	SB431542	-	1 µM	-	-	10 mM	-	500 nM	-	-	-	-	-
	A83-01	500nM	500 nM	500 nM	500nM	500 nM	500 nM	-	500 nM	0 or 50 ng/mL	-	5 μM	-
FGF	FGF2	1 ng/mL	-	-	10 ng/mL	-	-	-	-	20 ng/mL	20 ng/mL	-	10 or 100 ng/mL
	FGF7	-	100 ng/mL	25 ng/mL	-	25 ng/mL	-	25 ng/mL	25 ng/mL	-	-	-	-
	FGF10	20 ng/mL	100 ng/mL	100 ng/mL	10 ng/mL	20 ng/mL	100 ng/mL	100 ng/mL	100 ng/mL	-	-	-	-
EGF	EGF	50 ng/mL	-	-	50 ng/mL	-	50 ng/mL	-	-	50 ng/mL	50 ng/mL	50 ng/mL	10 or 100 ng/mL
p38 MAPK	SB202190	10 μM	-	1 μM	5 μM	10 mM	-	500 uM	500 nM	-	-	3 μM	-
ROCK	Y-27632	-	5 µM	5 μM	10 μM	10 mM	10 μM	5 uM	10 μM	10 μM	10 μM	10 μM	-
etc		1 μM PGE2, 10 nM Gastrin 1	0 or 10 µM Nutlin-3a, 40 ng/mL Heregulin β-1	5 μM Nutlin-3a	1 μM PGE2, 20 ng/mL HGF	-	100 ng/mL FGF4, 100 nM SAG	-	-	-	-	10 μM Forskolin, 3 nM Dexamethasone	10 or 100 ng/mL NRG1, 10 or 100 ng/mL IGF1, 10 or 100 ng/mL ActivinA, 10 μg/mL transferrin
supplement	NA	10 mM	10 mM	10 mM	-	10 mM	-	-	5 mM	-	-	5 mM	-
	B27	1×	1×	1×	1×	1×	1×	1×	1×	1×	1×	2% (*v/v*)	-
	N2	1×	-	-	1×	-	-	-	-	1×	1×	1% (*v/v*)	-
	NAC	1 mM	1.25 mM	1.25 mM	4 mM	1.25 mM	1.25 mM	-	1.25 mM	-	-	1 mM	-
	etc	-	-	-	-	-	-	-	-	-	-	-	1× trace elements A, B, C 1× nonessential amino acids 50 μg/mL ascorbic acid
Base medium		Ad-DF+++	DMEM/F12	Ad-DF+++	Ad-DF+++	Ad-DF+++	Ad-DF+++	Ad-DF+++	Ad-DF+++	Ad-DF+++	DMEM/F12	DMEM/F12	DMEM/F12
matrix		Matrigel (ratio is not determined)	100% Matrigel	10 mg/mL geltrex	100% Matrigel	100% Matrigel	100% matrigel	10 mg/mL Cultrex	100% matrigel	66% matrigel	66% matrigel	100% Matrigel	100% Matrigel
success rate		71%	-	41%	55%	80%	63% (short-term) 10% (Long-term)	88%	83%	80%	56%	79%	80%
resected source		NSCLC (*n* = 14)	AC (*n* = 6)	Total (*n* = 59) AC (*n* = 46) SCC (*n* = 4) LCNEC (*n* = 2) NSCLC (*n* = 7)	Total (*n* = 11) AC (*n* = 10) ASC (*n* = 1)	AC (*n* = 15)	Total (*n* = 30) AC (*n* = 16) SCC (*n* = 14)	NSCLC (*n* = 16)	advanced AC (*n* = 100)	SCLC (*n* = 10)	Total (*n* = 36) AC (*n* = 23) SCC (*n* = 8) LCC (*n* = 2) SCC (*n* = 2) ASC (*n* = 1)	Total (*n* = 103) AC (*n* = 71) SCC (*n* = 23) SCLC (*n* = 4) other (*n* = 5)	Total (*n* = 125) AC (*n* = 82) ASC (*n* = 6) SCC (*n* = 31) LCC (*n* = 4) Pleomorphic (*n* = 2)

AC: adenocarcinoma, SCLC: small-cell lung cancer, NSCLC: non-small-cell lung cancer, SCC: squamous cell carcinoma, ASC: adenosquamous cell carcinoma, LCC: large-cell carcinoma, LCNEC: large-cell neuroendocrine carcinoma.

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
