# Peer review of "The Patient-Derived Cancer Organoids: Promises and Challenges as Platforms for Cancer Discovery"

_cancers, 2022, doi:10.3390/cancers14092144_

Round 1

Reviewer 1 Report

This review is well-written and valuable for the researchers of cancer biology or tissue engineering.

However, there are some concerns that remained. Some sentences or descriptions should be added for the revision. The paper would be re-considered only when all the comments were responded and reflected in a revised version.

The reviewer’ comments are below.

1.

Between sections 2.2 and 2.3

The biomaterial technologies for the support of 3D culture models should be introduced briefly by quoting related reviews and research papers. The biomaterial can support the ECM components and assist the characteristics of 3D models. In the future, patient-derived cancer organoids will be combined with biomaterials.

For example,

Overall for concept    Cancers 202012(10), 2754

Alginate  Biomaterials 55 (2015) 110-118

Chitosan  Biomaterials 25 (2004) 5147–5154

Gelatin  Tissue Eng. Part C Methods 201925, 711–720 https://doi.org/10.1089/ten.tec.2019.0189

Collagen  doi.org/10.1016/j.actbio.2018.06.003

Hyaluronic acid  Adv. Healthcare Mater.2015, 4, 1664–1674

  1. Figure 1

The standard for the determination is not clear.

  1. Overall

Recent results should be summarized in Table.

  1.  

Before the conclusion section, future perspectives should be added because of the review.

Reviewer 2 Report

The authors have put together a very well-organized manuscript on the patient derived cancer organoids as the future of cancer research. This subject is of utmost importance and would definitely bring interest to fellow researchers, however, my only concern at this point is the availability of similar review articles in the past year focusing on patient derived organoid technology for cancer and I am not able to differentiate this paper from the rest. Therefore, I would request the authors to re-shape the manuscript in such a manner to attract more readership and to differentiate it from already available reviews.

Some quick examples of reviews available include:

1) Pernik, Mark N., et al. "Patient-derived cancer organoids for precision oncology treatment." Journal of Personalized Medicine 11.5 (2021): 423.

2) Wensink, G.E., Elias, S.G., Mullenders, J. et al. Patient-derived organoids as a predictive biomarker for treatment response in cancer patients. npj Precis. Onc. 5, 30 (2021). https://doi.org/10.1038/s41698-021-00168-1.

As a suggestion, the authors could bring in more in-depth literature to the PDCOs developed for different types and subtypes of cancers including their specific challenges and promises. A more detailed review is requested before acception for publication.

Reviewer 3 Report

In the review article entitled “The patient-derived cancer organoids: Promises and challenges as platforms for cancer discovery”, the authors discussed recent advances in PDO development in general. The schematic representations were drawn very well and conveyed the point. A few points to be noted:

  1. The review article became extremely general than scientific discussion. The authors did not discuss the problems faced to establish organoids like various media compositions. They should talk about various media compositions being used and how altering the composition helps in the success rate.
  2. The success rate is highly varying between each tumor and the authors also should suggest any modification if possible, to increase the success rate.
  3. There are some grammatical errors and please correct them.

Round 2

Reviewer 1 Report

The manuscript has been significantly improved.

I recommend the publication.

Reviewer 2 Report

The authors have implemented the suggestions and changes provided. I accept the manuscript for publication in its current form.